# OpenReview forum: "Microsaccade Inspired Probing: Positional Encoding Perturbations Reveal LLM Misbehaviors"
_ICLR.cc/2026/Conference — ICLR 2026 Conference Withdrawn Submission_

### Official Review · Reviewer_vRFo · 2025-10-27

**Soundness:** 3
**Presentation:** 3
**Contribution:** 3
**Rating:** 6
**Confidence:** 4

**Summary:**

This paper investigates whether large language models internally encode signals indicative of their own anomalous behaviors. Inspired by microsaccades in visual neuroscience, the authors propose Microsaccade-Inspired Probing (MIP), which introduces lightweight sinusoidal perturbations to positional encodings and measures resulting shifts in attention and next-token distributions to detect anomalies. A lightweight MLP classifier is trained on these difference features. Experiments on multiple Llama and Qwen models show near-perfect AUC on jailbreak and backdoor detection, improvements on factuality, and moderate gains on toxicity. The method is computationally efficient (O(1)) and suitable for real-time monitoring.

**Strengths:**

1.The paper is well-format and easy to read.
2.The method is elegant and highly efficient, requiring no fine-tuning or architectural changes while achieving constant-time complexity O(1) through interventions limited to the positional encoding layer.

**Weaknesses:**

1.The paper lacks a released code repository, which limits reproducibility.
2.Section 2 closely resembles the writing style and structure of LLMScan and should be substantially revised.
3.Please provide the explicit formula of PE^MIP to avoid confusion. If this function includes hyperparameters, their values and impact on performance should be clearly reported.
4.In lines 252–260, aside from factuality and jailbreak/backdoor tasks, is there any heuristic explanation for the relationship between positional encoding and toxicity detection? The evaluation performance appears notably strong and deserves further interpretation.
5.Experiments should include a wider range of LLM architectures (e.g., DeepSeek, Mistral) to strengthen generalizability.
6.The format in Table 1 for baseline and ‘ours’ should be consistent; the use of “–” in baselines is unconventional.
7.It is unclear why comparisons with LLMScan are limited to layer causality only rather than the full baseline (layer + token causality).
8.The section “Challenges in Toxicity Detection” lacks the corresponding key result figures.
9.PCA and LDA results show weaker separability for Sleeper compared to WikiData, yet the main results differ, this discrepancy requires explanation.

**Questions:**

1. Please provide the explicit formula of PE^MIP to avoid confusion. If this function includes hyperparameters, their values and impact on performance should be clearly reported.
2. Is there any heuristic explanation for the relationship between positional encoding and toxicity detection?
3. Compare with full version of LLMScan.
4. Further explanation for PCA and LDA results.

---

### Official Review · Reviewer_DMse · 2025-10-30

**Soundness:** 1
**Presentation:** 1
**Contribution:** 1
**Rating:** 0
**Confidence:** 4

**Summary:**

The paper introduces a method (MIP) for runtime detection of various LLM misbehaviors (such as false statements, toxic outputs, or jailbreaks) using a supervised classifier. MIP runs a second forward pass of the LLM, but with a perturbation vector added to the initial embeddings. Then it compares the attention patterns between the original and the perturbed forward pass and computes their L2 distance for each attention head and layer. These L2 distances are finally used as input features to a small MLP trained as a supervised classifier.
The perturbation vector added to the embeddings is a sinusoidal position embedding vector.
The paper has experiments on factuality, jailbreaks, toxicity, and backdoors, where it beats a previous method, LLMScan.

**Strengths:**

The topic is important, and feature engineering based on LLM activations is an interesting direction. The new MIP method beats LLMScan on a wide range of datasets while being computationally cheaper.

**Weaknesses:**

The paper essentially does not motivate any of the complexity of MIP and does not demonstrate empirically that this added complexity helps. For example, it's unclear what motivates adding sinusoidal embeddings, and in fact, Fig. 6 in the appendix shows that using random embeddings works almost as well, with no statistically significant differences in most models. Even more importantly, the paper does not ablate whether the feature engineering as a whole helps compared to using raw activations as input to the classifier (or other simpler features). Given that MIP doubles inference costs by requiring a second forward pass, checking whether this helps compared to more straightforward probes seems like a basic and crucial ablation.

The paper tries to motivate MIP by drawing tenuous connections to microsaccades in human vision, but I found that aspect distracting rather than illuminating. I also found the presentation difficult to follow in other places; the paper spends a lot of space on basics such as how language modeling works (end of section 3) but becomes more sparse and confusing when describing the method itself (section 4). For example, the term PE^MIP(TE(d)) from the key equation describing the perturbation isn't clearly defined (PE^MIP seems to just be the sinusoidal embedding formula based, but then why does it get the token embeddings TE as an argument?) I ended up looking through some of the code just to confirm what exactly the perturbation is.

MIP does seem to beat the one baseline the paper compares to, LLMScan. And sufficiently strong empirical evidence can of course overcome a priori skepticism about a method. But in this instance, I'm unconvinced for two reasons:
- Using a single baseline and only a small set of ablations, combined with the fact that those ablations actually suggest some parts of MIP don't contribute much, does not meet this bar of evidence in my view.
- While the LLMScan paper claims to beat more well-established baselines, at least some of the methods it compares to do not use supervised data. E.g. in the backdoor setting, LLMScan (and thus transitively MIP) compare against only ONION. ONION is an outlier detection method that does not need to be trained on labeled harmful vs benign data. MIP trains a supervised classifier, which is a much much easier problem than what ONION tries to solve. (Even linear probes on raw activations can often perform very well at backdoor detection when trained on supervised data.)

As a final note, the paper describes a "Gaussian noise" baseline, but the code in `_get_gaussian_positional_encoding_intervention` doesn't use any noise, it places a Gaussian PDF over the sequence dimension and then simply repeats this along the embedding dimension. (The "random" baseline does use actual Gaussian noise.) I didn't read the entire code base, so of course it's possible that I'm misunderstanding how this code is used, but as of now it doesn't inspire confidence in the overall soundness of the results.

**Questions:**

Am I meaningfully misunderstanding how MIP works (e.g. that it uses supervised training data)?
Are there concrete a priori reasons to think that MIP should outperform a classifier that simply uses the raw activations of the model (or some dimensionality reduced version) as features?

---

### Official Review · Reviewer_zast · 2025-10-31

**Soundness:** 1
**Presentation:** 1
**Contribution:** 1
**Rating:** 2
**Confidence:** 3

**Summary:**

This paper reports that lightweight position encoding perturbations appear to elicit latent signals associated with model misbehaviour. Without fine-tuning or task-specific supervision, the proposed method is empirically shown to detect diverse failures, including issues of factuality, safety, toxicity, and backdoor attacks, across LLMs.

**Strengths:**

The paper evaluates the proposed method across multiple datasets and LLMs, providing empirical evidence of its effectiveness.

**Weaknesses:**

- Limited Novelty and Contribution:
The proposed method appears to be a minor variation of the baseline (LLMScan), with the primary modification being the replacement of the detector with an MLP. This modification does not lead to a significant performance improvement. The main potential contribution (increased efficiency) is neither emphasized nor thoroughly analyzed.

- Lack of Theoretical Soundness:
The paper provides no mathematical derivation or theoretical justification for its central claim. The methodology (Sections 3 and 4) relies on pseudo-mathematical notations rather than properly defined formulations, making the exposition unclear.

- Flawed Experimental Setup:
    1. The paper includes only a single baseline (LLMScan), and the reported results for this baseline differ substantially from those in the original LLMScan paper.
    2. The choice of datasets (e.g., for the Toxicity task) and LLMs differs from those used in the baseline work without clear justification.

- Inconsistent Results and Poor Presentation:
    1. Numerical Inconsistency: The numerical results presented in Figure 2(b) and Table 1 are highly inconsistent, raising serious concerns about the correctness of the results or labeling.
    2. Misleading Figures/Tables: The presentation of Table 1 may misleadingly suggest baseline accuracy is variable. The colors in Figure 2 are difficult to distinguish. Figures 2 and 3 omit results for one dataset mentioned in the text, and this discrepancy is not explained.

**Questions:**

1. Could you please clarify the stark numerical inconsistency between Figure 2(b) and Table 1? This seems to be a critical error.
2. Why do your reported results for the LLMScan baseline differ substantially from those presented in the original LLMScan paper?
3. Please justify the decision to use different datasets (especially for the Toxicity task) and LLMs compared to the baseline study.
4. Given the methodological similarity to LLMScan, could you (a) clearly articulate the core novelty of your method beyond the MLP detector, and (b) provide a quantitative analysis of the efficiency gains, as this seems to be the main potential contribution?

---

### Official Review · Reviewer_fm8L · 2025-11-01

**Soundness:** 2
**Presentation:** 2
**Contribution:** 2
**Rating:** 2
**Confidence:** 4

**Summary:**

The authors propose MIP, a way for training classifiers on the change in the model's next-token distributions and attention maps after perturbing the positional embeddings. Specifically, they amplify the positional encoding tokens and construct features based on the distance between this forward pass and a normal forward pass. They then feed these features to an MLP and train it to classify toxicity, jailbreaks, backdoors and factuality in the model's responses. They compare it with another probing method, LLMScan, and show that MIP outperforms it in most cases.

**Strengths:**

1. Provides a unique way of probing language models, which outperforms a previous causal probing method.
2. The paper reports its results on a variety of datasets, all of which are relevant to AI safety.
3. PCA and supervised LDA plots show that the features crafted by the authors separate into clusters for each dataset considered.

**Weaknesses:**

The paper requires many additional baseline comparisons. The literature review and baselines do not consider classifiers acting directly on the model's residual stream (see [1, 2, 3, 4], for example). These probes are one of the most popular classes of probes in the literature.


[1] Goldowsky-Dill, Nicholas, Bilal Chughtai, Stefan Heimersheim, and Marius Hobbhahn. "Detecting Strategic Deception Using Linear Probes." arXiv preprint arXiv:2502.03407 (2025).

[2] Anthropic: Simple Probes Catch Sleeper Agents. https://www.anthropic.com/research/probes-catch-sleeper-agents

[3] McKenzie, Alex, Urja Pawar, Phil Blandfort, William Bankes, David Krueger, Ekdeep Singh Lubana, and Dmitrii Krasheninnikov. "Detecting High-Stakes Interactions with Activation Probes." arXiv preprint arXiv:2506.10805 (2025).

[4] Cunningham et al. Cost-Effective Constitutional Classifiers via Representation Re-use (https://alignment.anthropic.com/2025/cheap-monitors/)

**Questions:**

1. Why don't they compare their method with activation probes as described under point 1 of weaknesses? Could this be added?
2. Regarding details of the probe training dataset:
    - Consider Factuality: It is tested on 3 datasets. Is a different probe trained on each dataset?
    - If yes, it would be very useful to know if the probes generalise when trained on only one of the datasets (or if they generalise to some other significant distributional shift), as out-of-distribution generalisation is very important for safety monitors.
3. In Appendix G, what is the intuition behind perturbing positional encodings with noise? It would also be helpful if this could be drawn with the baseline perturbation used in the main body.

Regarding writing:
- In the abstract: I would recommend replacing 'state-of-the-art LLMs' with open-sourced models with the parameter range of the models.
- Could the related work please be updated with literature on activation monitors?

---

### Note · Authors · 2025-11-13

I have read and agree with the venue's withdrawal policy on behalf of myself and my co-authors.